# Expected length of stay in residential aged care facilities in Australia: Assessing the impact of dementia using machine learning

Alexander Fracalossi[1], Jinhui Zhang[1], Evelyn Lee[2]*, Yanlin Shi[1]

1 Department of Actuarial Studies and Business Analytics, Macquarie Business School, Macquarie University, New South Wales, Australia, 2 Leeder Centre for Health Policy, Economics & Data (The Leeder Centre), University of Sydney, New South Wales, Australia

* evelyn.lee@sydney.edu.au

## 1. Introduction

Australia's population is ageing at an unprecedented rate [1]. According to the Australian Bureau of Statistics, the percentage of Australians aged 65 and over has increased by almost fourfold over the last decade to 2021 (4% in 2010 to 16.2% in 2021) and it is projected to reach 20.1% by 2041 [2].

Population ageing has several implications for the healthcare system and society. While a longer life means individuals have more years of living, an ageing population is associated with increased number of people with multimorbidity, a diagnosis of dementia and living with disability that would likely put significant burden on the broader health system and aged care services [3]. Indeed, it is estimated that the total health spending for older population would reach A$320 billion in 2035 and constituted 8% of the Australian economy [4].

There is a growing concern whether the current health systems and community care are equipped to provide quality care to older population in an equitable and cost-effective way. Thus, over the recent years, several reforms to the Australian aged care system have been undertaken to become more consumer-focused where older people are provided with choice and control over what type of out-of-hospital programs service they wish to receive – in their home, community-based care or residential aged care (RAC) settings [5]. This reform is also coupled with significant funding reform that redirect previous block-funding of selected providers to home care packages to encourage more older people to stay at home for longer as was done in other countries such as the United Kingdom and Canada [6,7].

Snice the reform, by 2021, it is estimated that more than a million of older people were accessing a home care package due to a greater preference of living independently at their own home [8]. The publicly funded home care packages are rationed according to needs and financial means. Consumers can access to a broad and coordinated mix of services, e.g., home help and personal care, physiotherapy, meals, transport, nursing, allied health, and day care based on their assessed needs.

However, studies have found that despite the government efforts to create a more balance aged care system with home care services as an alternative to the high-cost RAC, almost 80% of individuals who have previously used home care services transit

**Data availability statement:** This paper uses publicly available unit record de-identified data from the residential aged care data provided by the AIHW, sourced from the National Aged Care Data Clearinghouse. Permission to use this dataset was granted by the AIHW Ethics Committee. Data cannot be shared publicly because of confidentiality. Data are available from the AIHW Ethics Committee (contact via AIHW website) for researchers who meet the criteria for access to confidential data. Source: Institute of Health and Welfare (AIHW) GEN Aged Care Data (Source: https://www.gen-aged-caredata.gov.au/resources/access-data/2017/august/gen-data and https://www.gen-aged-caredata.gov.au/).

**Funding:** The author(s) received no specific funding for this work.

**Competing interests:** The authors have declared that no competing interests exist.

to RAC when their functional (physical and/or mental) capacity declines that impact their day-to-day functioning [9,10]. Data from the Australian Institute of Health and Welfare (AIHW) indicated that the majority of people who entered permanent RAC were frail and suffered from multiple health conditions that requires continual and substantial care assistance [11,12]. The majority of users were aged between 65 and 74 as well as those over 90 with higher dependency level [5]. Although RAC have adequate access to full time clinical and personal care and support, the increasing rates of residents with extensive care requirements for complex chronic conditions (e.g., dementia, mental health disorders) and multimorbidity entering RAC would place significant cost burden to the aged care system. Indeed, between 2008-9 and 2018-19, the rate of Commonwealth recurrent expenditure was rising at a faster rate (7% per year) than the 1.3% increase in the rate of residents entering RAC which is a marked turnaround from the intended funding reform [14,15,16].

Under the current financing arrangements, expenditure for aged care services is funded by the federal government with co-contribution from non-government organisations and consumers depending on their financial circumstances [8].The Aged Care Funding Instrument (ACFI) is used as a funding tool to assess the needs of individuals on admission to RAC or when need changes to determine the level of care payment from the government to the providers [13]. The ACFI consists of 12 care need questions framed within three categories based on 3 domains- Activities of daily living (ADL), behaviour (BEH) and complex health care (CHC). The highest funding was allocated to residents with higher dependency needs, such as those who are bedridden or have severe mental illness) based on the assessment.

In large part, the ACFI was introduced as a means to contain costs by redirecting funding towards residents with higher dependency care needs [17]. However, previous estimate showed that almost two-thirds of the total Commonwealth funding for aged care services (A\$13.0 billion in 2018-19) were spent on one-fifth of the total aged care consumers (21%) on residential care subsidies and supplements, and accommodation. Cost containment has become one of the most critical challenges for the aged care system.

Previous analyses suggest that the funding growth was largely driven by the increasing number of people with high dependency and care needs on admission [5]. Other studies have found that demographics trend (e.g., age, gender, marital status, financial status), disease patterns (e.g., dementia, diabetes, heart problems) and aged care workforce may be contributing to the higher costs [18,19,20,21]. However, most of these studies do not take into account the effect of dependency rating and length-of-stay that underpinned the increased expenditure.

Moreover, previous analyses were conducted using standard statistical modelling, e.g., regression analyses which are based on theoretical assumptions and subject knowledge for model specifications and control for confounding variables [19,20,21]. More recently, there has been a growing interest in using machine learning (ML) methods as an alternative approach to inform patient care and improve the efficiency of health-care delivery [18]. Unlike the traditional statistical approach, ML does not assume a priori knowledge about the statistical distributions and thus, is able to

process complex relationship between variables that may not have been previously identified using conventional techniques [22].

As the demand for aged care services are expected to increase with ageing population, identifying factors that influence RAC use within these populations is important to better inform resource allocation decisions. This present study aims to develop a model to generate new insights into individual-level factors predicting RAC use. Additionally, we aim to compare machine learning techniques along with standard logistic regression to determine which model performs best in predicting RAC service use (in terms of duration of stay) based on a large nationally representative data.

## 2. Methods

### 2.1 Data

The data for this study was approved and obtained from the Australian Institute of Health and Welfare (AIHW) GEN Aged Care Data sourced from the National Aged Care Data Clearinghouse on 19 December 2022. The data contains 793,323 records of all Australian RAC residents admitted into aged-care facilities from 2008–2009 to 2018–2019. The GEN data included de-identified information on the residents such as age range (five-year age groups), gender, marital status, admission date and discharge, dementia status, and service locations (state and territories). While the data contains records for both permanent and respite aged care residents, permanent aged care data were used in our analyses. Data privacy and confidentiality were ensured to fulfil the requirements of the Australian Institute of Health and Welfare Act 1987. Permission to use this dataset was granted by the AIHW Ethics Committee.

In addition to the AIHW dataset, the data was also linked to the Aged Care Funding Instrument (AFCI) scores that assessed the residents' care needs across a range of ADL, BEH and CHC domains (S1 Table). Prior to admission to the aged care, all residents were assessed on their needs level and assistance using AFCI score. The scores are then summed and used to categorise the resident's needs into 4 categories – Nil, Low, Medium or High. The higher the score in each domain (i.e., ADL, BEH or CHC) the higher the daily ACFI subsidy from the government. For example, the daily rate for a resident with high needs for the activities of daily living domain according to the ACFI score was A$115.49 in 2020-2021 [23].

**Outcome variable.** The length-of-stay (LOS) at the aged care facility as measured in days was the outcome of interest. To avoid the influence of outliers, LOS corresponding to the smallest 2.5% (< 3 months) and largest 2.5% (>94 months) of LOS were removed which represented 36% of the total records resulting in a final 509,142 observations remained. To be comparable with other studies, LOS was measured in days [19,24].

### 2.2 Statistical analyses

The purpose of using ML techniques is to produce a model which could identify as accurately as possible the predictors of RAC use and stay duration which are important for policy makers when making resource allocation decisions. The current ensemble of ML models –Random Forest, Gradient Boosting Decision Trees (GBDT) and traditional regression method were selected based on their rigorous algorithms that are able to support decision makers in identifying complex patterns of RAC use and the duration of stay.

**Linear regression.** We use the multiple linear regression to investigate the relationship between LOS (outcome of interest) and its associate covariates. Multiple linear regression constitutes a statistical methodology employed for the purpose of estimating the intricate dynamics existing between an outcome and predictor variables. Fundamental to this approach is the presumption of a linear association between the outcome and each individual predictor variable in the model. It is acknowledged that, in real-world scenarios, this linearity assumption may not invariably hold; nevertheless, its adoption facilitates a straightforward model fitting process while still affording a degree of explicatory efficacy. To preliminarily examine the influences of concerned explanatory variables on LOS, we employ the multivariate linear regression.

To investigate the potential non-linear relationship between variables, a basis spline function was employed to model the age, year of admission, ADL score, and BEH score. Six degrees of freedom is selected to optimise the out-of-sample fitting performance (as measured by the root of mean squared error, or RMSE). The resulting linear regression model included demographics (age groups, gender), dementia status, ADL score, BEH score, marital status, CHC level, admission year and state and territories.

**Random forest.**  Random forest (RF) is a learning method for classification by generating a large number of decision trees built on bootstrap samples from the dataset. It randomly splits structures the data akin to a tree, commencing with a 'root node' encompassing all data, which is then divided based on a chosen splitting criterion, often involving variance reduction. The algorithm recurrently forms branches embodying decisions rooted in feature values, halting when predefined stopping conditions like maximum tree depth or minimum node size are met. Each terminal node or 'leaf' encapsulates an output prediction. The random forest model amalgamates numerous trees to yield a final prediction. The process involves averaging across decision trees, enhancing the model's generality and robustness. More importantly, compared to a linear regression, random forest is able to accommodate (unknown) non-linear influence of explanatory variable on the LOS, which will be discussed in Section 3.1. As such, our approach incorporates a random forest model, diverging from using a lone decision tree, to forecast LOS.

A random forest model can be described mathematically as

$$\hat{y} = \frac{1}{B} \sum_{b=1}^{B} T_b(x) \tag{1}$$

where ŷ represents the predicted target variable, B is the number of trees in the random forest, and T_b (x) represents the prediction of the b-th tree for the input features x. As discussed earlier, this is equivalent to the random forest averaging the predictions of B different decision trees for a given data point.

**Training and validation.**  In training this model, we employed a cross-validation approach to optimise the initial selection of features used for data splitting. Cross-validation is a resampling technique that assesses predictive models by dividing the original sample into a training set for model training and a test set for evaluation. This technique guards against overfitting by providing a robust assessment of model performance on unseen data, thus aiding in model selection.

In k-fold cross-validation, the data is randomly shuffled and divided into k groups or "folds". Each distinct group is treated as a test dataset, while the remaining groups form the training dataset. The model is trained on the training set and validated on the test set. This process is iterated k times, using each of the k groups as validation data once. The k results are then averaged to generate an unbiased estimate of model prediction performance. For this analysis, we chose k=3, striking a balance between robust results and computational efficiency.

**Gradient boosting.**  Gradient boosting is an alternate machine learning algorithm which is used for capturing non-linear relationships in data sets. It, like a random forest model, utilises decision trees to make a series of splits on the data that are subsequently used for prediction. However, unlike the random forest model, a gradient boost model uses boosting.

Boosting is a method of ensemble learning – which is a model that makes predictions based on the amalgamation of several simpler models. Boosting involves fitting a series of simple decision trees which have one split, with each tree learning from the mistakes of the previous model. Models are added sequentially until no further improvements can be made.

Gradient boosting is particularly useful for capturing non-linear relationships in data It does this by combining multiple decision trees to create a final model that can handle complex relationships between predictors and the target variable. Each decision tree tries to correct the residuals errors left by the previous tree.

## 3. Results

### 3.1 Sample characteristics

A total of 509,142 residents were included in the analysis. Over the study period, more than half of the aged care residents were female (54.6%), with three-quarter of them aged 80 years old and over (77.6%). More than half of the residents lived with dementia although these rates have fallen by 7 percentage points between 2008-9 and 2018-19. Conversely, the overall number of people assessed with higher complex needs has increased from 46.6% in 2008-9 to 65.6%, representing 11,711 in RAC settings (Table 1).

Our analyses showed that women stayed in RAC longer compared to their male counterparts (210 more days). Similarly, those with dementia stayed ~200 more days in RAC compared to those without which suggests the positive association between LOS and levels of complex health care.

In Fig 1, we showed four scatter plots of a sub-dataset consisting of 5,000 observations randomly sampled from our dataset. In Figs 1-4, the yellow line indicating the smoothed curve fitting of the pronounced relationship between the LOS and Age, Admission Year, ADL Score, and BEH Score, respectively. The curves are fitted using locally-weighted polynomial regression (LOWESS) based on the complete dataset. In all cases, we observe a potential non-linear relationship between LOS and each of the four continuous variables. These preliminarily support the employed splines for linear regression and ML models, as outlined in Section 2.2.

The relationship between LOS and ACFI were explored in Figs 5(ACFI-BEH score) and 6 (ACFI-ADL score) and 7 (ACFI-CHC). Our results showed a notable positive relationship between the ACFI-BEH (wandering, verbal and physical aggression, and depression) score and LOS and this is substantiated by their intermediate level. However, the relationship between the ACFI-ADL score and LOS is less clear suggesting a potentially non-linear dependency of LOS on ADL score. Finally, when we used the demographic pyramids to investigate distributions of residents across the different states and territories, age groups and gender, the study showed a disproportionately higher female aged care users compared with men, whereas the age-sex-specific distribution is consistent across different states and territories (Fig 8).

**Preliminary linear regression results.** Table 2 displays the estimated coefficients of the linear regression model, which was developed using a 70% training set and a 30% testing set. The results in Table 2 are based on the model trained with the training set and validated with the testing set. At 1% significance level, the linear regression model showed that predictive factors including gender, age, year of admission, dementia status, BEH score, marital status, and CHC level were significantly associated with LOS. The primary aim of using linear regression is to identify important factors for preliminary analysis before employing machine learning methods to build predictive models.

Although the results from linear regression have been shown to be robust for model assumptions, the model diagnostics found the presence of heteroscedasticity and non-normality which suggested that there was an insufficient linear dependency of LOS on the covariates. These issues persist despite modelling using a logged LOS.

**Random forest and gradient boosting results.** Unlike linear regression, the random forest model does not require such stringent assumptions. For the fitted random forest, the permutation importance of each variable is plotted in Fig 9, ranking from the highest to the lowest. The results showed that year of admission, ACFI-ADL score, and ACFI-BEH score were identified as strong predictors for LOS. This was followed by gender and dementia status. Notably, although ACFI-ADL score was not associated with LOS, it emerged as an important predictive factor in the random forest model, suggesting a non-linear relationship cannot be nested by the linear regression.

To cross validate the robustness of random forest results, we also fitted the gradient boosting model. The permutation importance is plotted in Fig 10 which showed similar results with random forest. Interestingly, both the ACFI-ADL score and ACFI-BEH score have non-trivial importance on influencing the LOS at an aged care facility.

The "importance" shown refers to permutation importance, a model-agnostic technique in machine learning for assessing the contribution of individual variables. It measures the effect of shuffling a variable on the model's performance. Specifically, the model's performance (e.g., RMSE) is first calculated on the original data, then recomputed after randomly

**Table 1. Characteristics of resident admitted to residential aged care services and Length of Stay (LOS) between 2008-9 and 2018-19.**

| Variable | Residential care | | | |
|---|---|---|---|---|
| | 2008-09 | LOS | 2018-19 | LOS |
| | N (%) | | N (%) | |
| All | 44740 | 925 | 17861 | 451 |
| | | | | |
| **Gender** | | | | |
| Male | 16968 (37.9%) | 770 | 8117 (45.4%) | 414 |
| Female | 27772 (62.1%) | 1020 | 9744 (54.6%) | 482 |
| | | | | |
| **Age group (years)** | | | | |
| 50-59 | 461 (1.0%) | 943 | 169 (0.9%) | 333 |
| 60-69 | 1890 (4.2%) | 947 | 797 (4.4%) | 431 |
| 70-79 | 8331 (18.6%) | 982 | 3037 (17.0%) | 434 |
| 80-89 | 24523 (54.8%) | 944 | 8329 (46.6%) | 462 |
| ≥ 90 | 9535 (21.3%) | 821 | 5529 (31.0%) | 452 |
| | | | | |
| **Marital status** | | | | |
| Married/partnered | 14737 (32.9%) | 826 | 7140 (40.0%) | 430 |
| Separated/divorced | 3176 (7.1%) | 959 | 1384 (7.8%) | 448 |
| Widowed | 23993 (53.6%) | 972 | 8007 (44.8%) | 473 |
| Never Married | 2834 (6.3%) | 1005 | 1330 (7.5%) | 432 |
| | | | | |
| **State** | | | | |
| Australian Capital Territory | 562 (1.3%) | 957 | 251 (1.4%) | 444 |
| New South Wales | 15253 (34.1%) | 917 | 5815 (32.6%) | 451 |
| Queensland | 7915 (17.7%) | 933 | 3298 (18.5%) | 450 |
| South Australia | 4333 (9.7%) | 923 | 1523 (8.5%) | 457 |
| Victoria | 11501 (25.7%) | 933 | 4804 (26.9%) | 450 |
| Tasmania | 1319 (3.0%) | 839 | 529 (3.0%) | 417 |
| Western Australia | 3758 (8.4%) | 945 | 1591 (8.9%) | 461 |
| Northern Territory | 99 (0.2%) | 972 | 50 (0.3%) | 507 |
| | | | | |
| Health condition-Living with dementia | 25,525 (57.1%) | 1014 | 9091 (50.9%) | 503 |
| | | | | |
| **ACFI score** | | | | |
| **Behaviour (BEH)** | | | | |
| Nil | 394 (7.6%) | 486 | 665 (3.7%) | 307 |
| Low | 6544 (14.6%) | 643 | 1904 (10.7%) | 354 |
| Medium | 10642 (23.8%) | 802 | 3946 (22.1%) | 412 |
| High | 24160 (54.0%) | 1117 | 11346 (63.5%) | 490 |
| | | | | |
| **Activities daily living (ADL)** | | | | |
| Nil | 492 (1.1%) | 534 | 34 (0.2%) | 299 |
| Low | 4778 (10.7%) | 607 | 654 (3.7%) | 371 |
| Medium | 10626 (23.8%) | 761 | 3109 (17.4%) | 418 |

*(Continued)*

**Table 1.** (Continued)

| Variable | Residential care | | | |
| --- | --- | --- | --- | --- |
| | **2008-09** | **LOS** | **2018-19** | **LOS** |
| High | 28844 (64.5%) | 1045 | 14064 (78.7%) | 463 |
| | | | | |
| **Complex health care (CHC)** | | | | |
| Nil | 2098 (4.7%) | 557 | 28 (0.2%) | 326 |
| Low | 9496 (21.2%) | 697 | 2013 (11.3%) | 385 |
| Medium | 12302 (27.5%) | 878 | 4109 (23.0%) | 448 |
| High | 20844 (46.6%) | 1094 | 11711 (65.6%) | 464 |

Abbreviations: LOS=Length of stay, ACFI=Aged Care Funding Instrument

The funding categories were based on 3 fundamental care needs domains- Activities of daily living (ADL), behaviour (BEH) and complex health care (CHC).

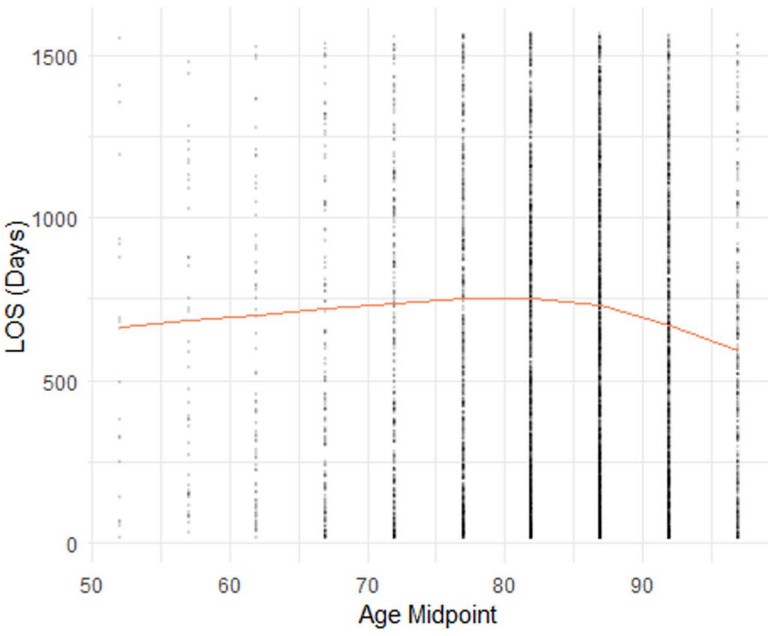

**Fig 1. Plot of Length of Stay (LOS) by Age group.**

permuting each variable. The importance is the difference between the original and permuted performance scores. A larger drop indicates a more critical variable for the model's predictions.

### 3.2 Comparison of ML techniques

To evaluate and compare the performance of linear regression and machine learning models (random forest and gradient boosting), we utilize two metrics: (pseudo) R-square and root mean square error (RMSE). The (pseudo) R-square metric is used only for in-sample comparisons and is interpretable primarily in the context of linear regression. RMSE, on the other hand, is used to assess out-of-sample prediction accuracy and is computed through 3-fold cross-validation, providing a robust measure of forecasting performance. The results, presented in S2 Table, show that both random forest and

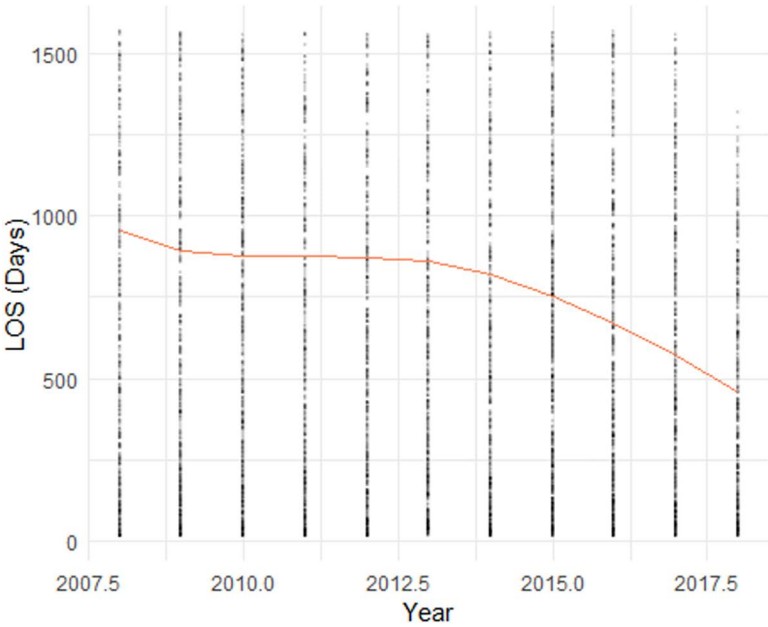

**Fig 2. Plot of Length of Stay (LOS) by Year of Admission.**

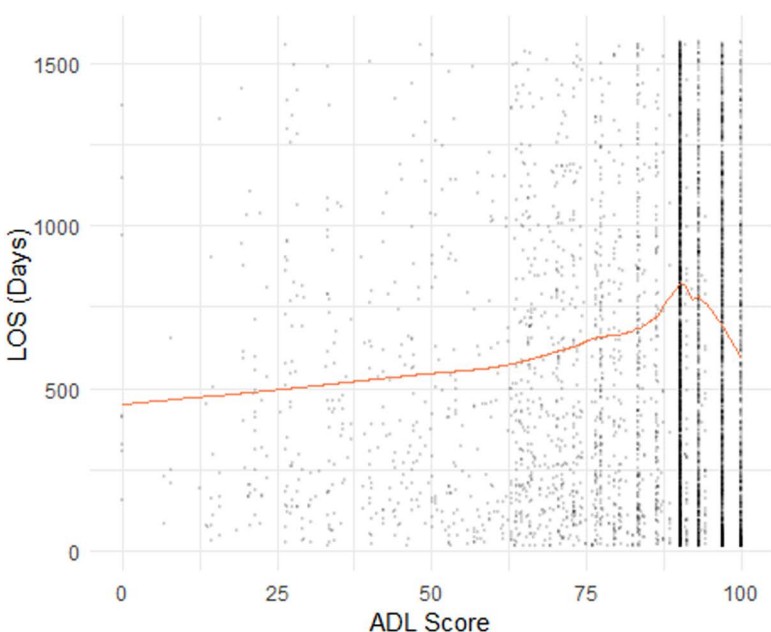

**Fig 3. Plot of Length of Stay (LOS) by Activities of daily living (ADL).**

gradient boosting models outperform linear regression. For in-sample performance, random forest and gradient boosting achieve higher (pseudo) R-square values (25% and 30%, respectively) compared to linear regression (16.8%). However, our final model selection prioritizes RMSE, as it reflects out-of-sample forecasting accuracy, which is the purpose of this

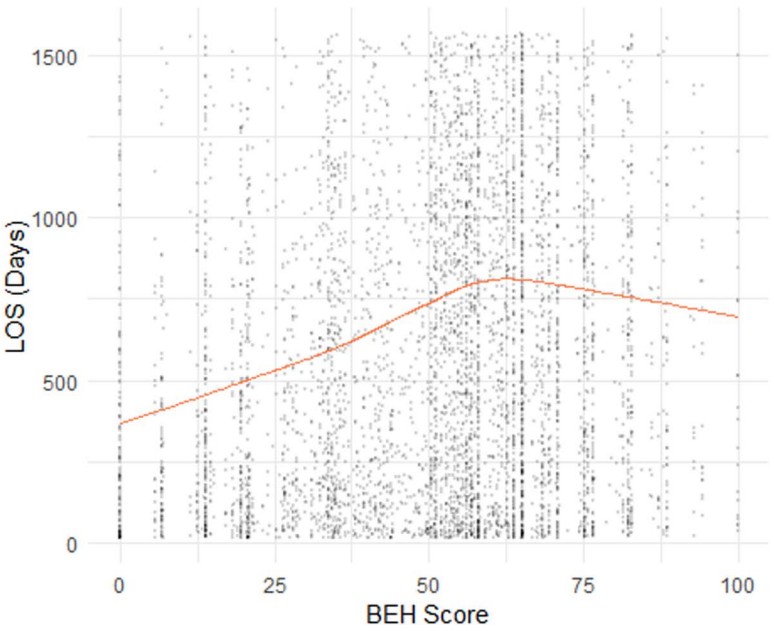

**Fig 4. Plot of Length of Stay (LOS) by Behaviour (BEH).**

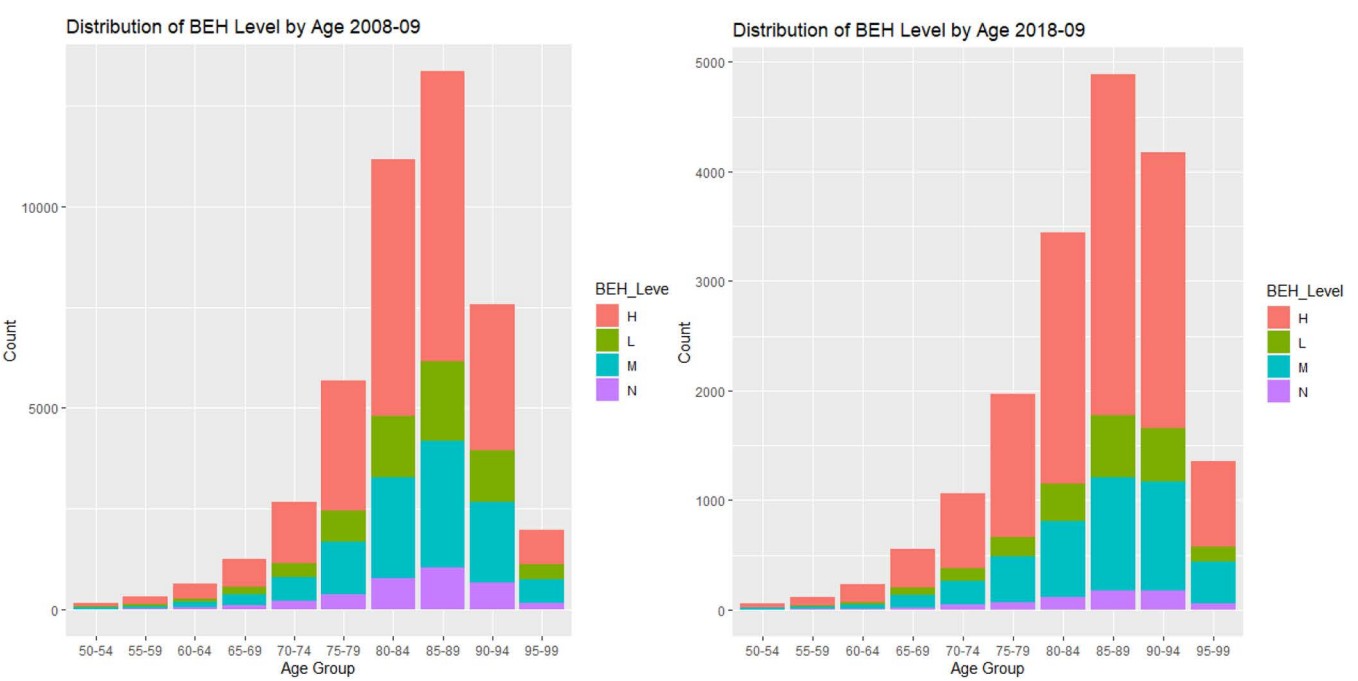

**Fig 5. Plot of Average Length of Stay (LOS) by Behaviour (BEH) between 2008-9 and 2019-20.**

paper. In this regard, random forest and gradient boosting demonstrate superior performance, with RMSE values approximately 6% and 7% lower than that of the linear regression model. Thus, the linear model is inferior to the two machine learning models.

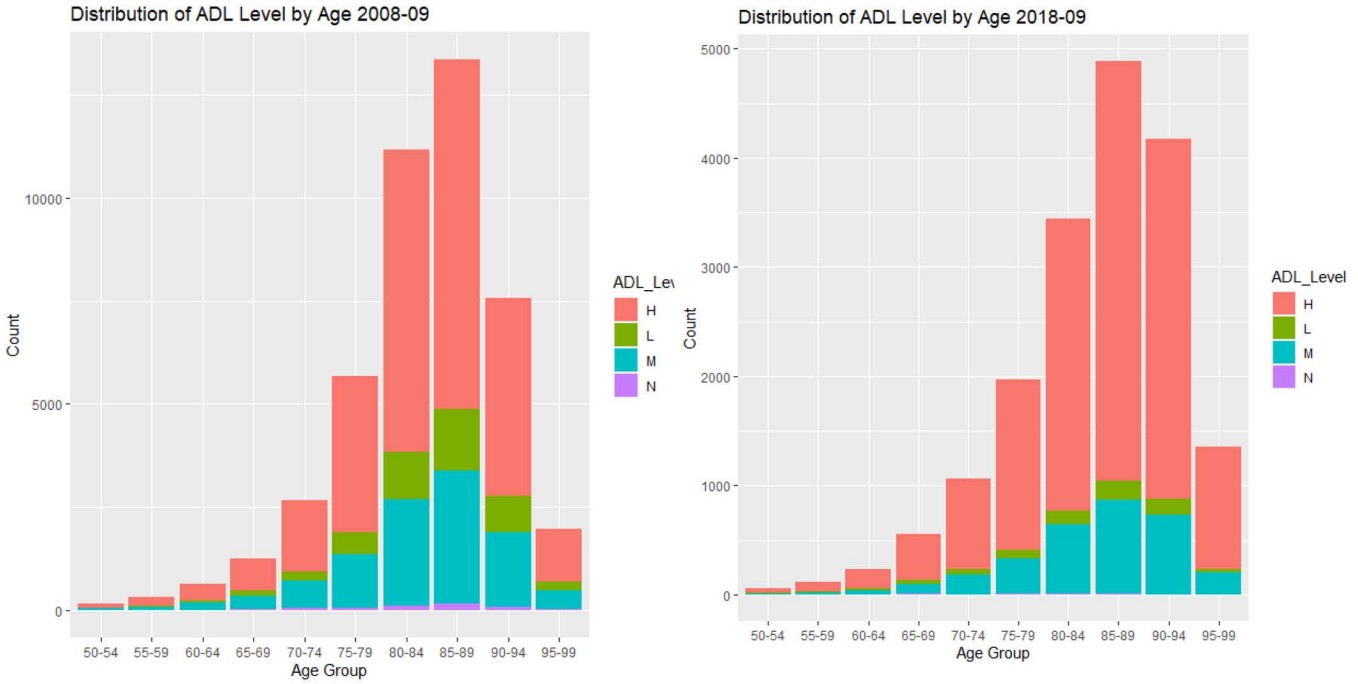

**Fig 6. Plot of Average Length of Stay (LOS) by Activities of daily living (ADL) between 2008-9 and 2019-20.**

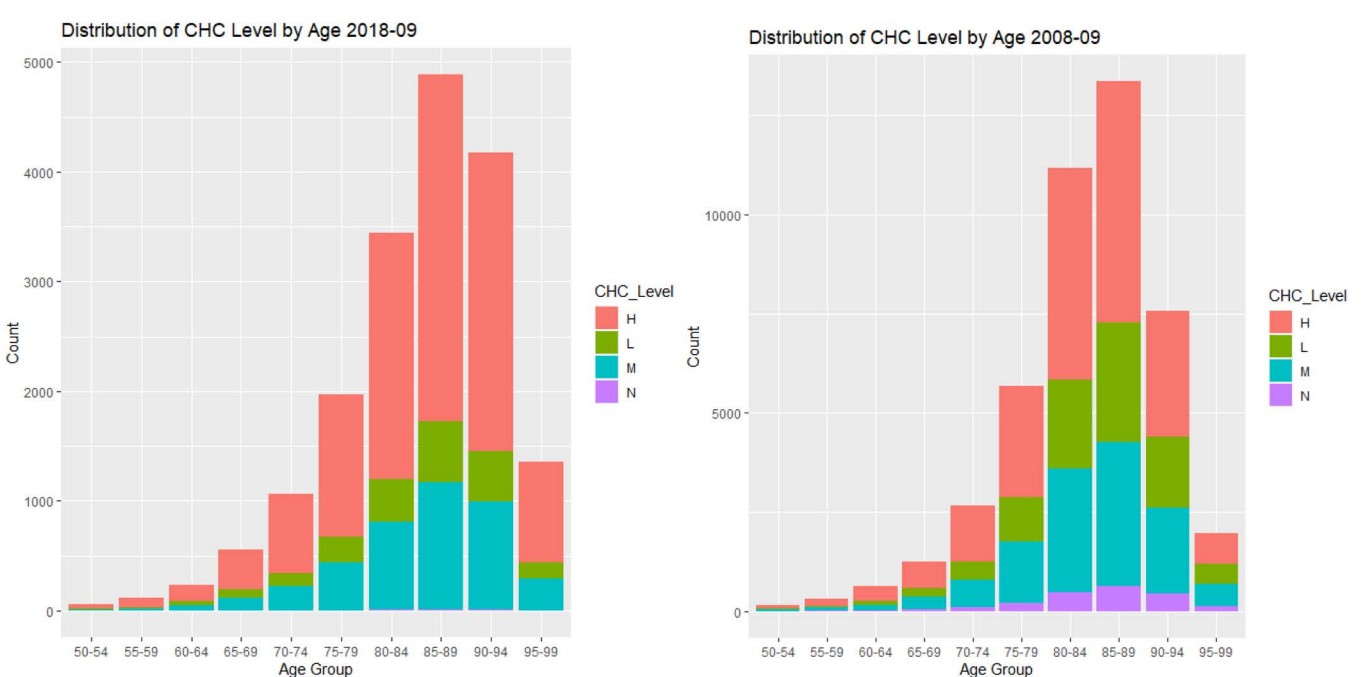

**Fig 7. Plot of Average Length of Stay (LOS) by Complex Health Care (CHC) between 2008-9 and 2019-20.**

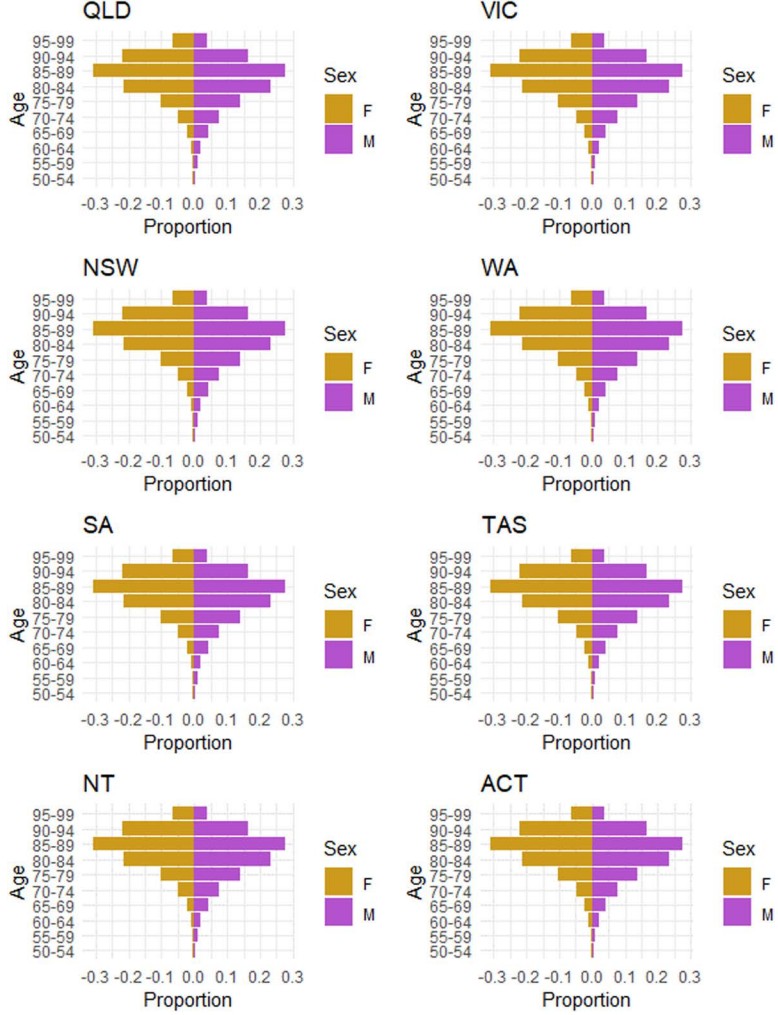

**Fig 8. Distributions in pyramid across states and territories.**

Our results showed that ML method using random forest and gradient boosting methods provided more robust findings compared with the traditional linear regression as it captured non-linear relationships and were less sensitive to changes or dependent on stringent assumptions.

### 3.3 Discussion

To our knowledge, this is the first study to use a longitudinal population-based linkage records of residents in RAC to predict their LOS in Australia. Our results are consistent with the literature that showed factors including gender, age, year of admission, dementia status, marital status, BEH score, and CHC level were significantly associated with LOS [18,20,24].

However, our study extends previous findings by providing evidence on the association between dementia status and LOS that will likely affect funding arrangements for aged care and resource allocation. For residents living or entering RAC, care needs are assessed by the ACFI which informed the cost of care to ensure efficient budget allocation. Indeed, a recent study showed that residents with dementia incurred an additional AU$9,100 annually per person to provide the additional care requirements [20]. However, other research found that the utility of ACFI failed to reflect quality of care for people with dementia as the assessments are applied in the same way as for other residents who do not live with the

**Table 2. Linear regression analysis predicting length of stay among people living in residential aged care.**

| | Estimate | Std. Error | t value | P-value |
|---|---|---|---|---|
| Intercept | 1273.886 | 51.787 | 24.599 | 0.000 |
| Age (years) | -4.023 | 0.442 | -9.099 | 0.000 |
| Being Male | -193.552 | 7.080 | -27.338 | 0.000 |
| Marital status | | | | |
| Married/partner | -58.025 | 13.886 | -4.179 | 0.000 |
| Never Married | 75.442 | 17.669 | 4.270 | 0.000 |
| Separated | 14.846 | 29.729 | 0.499 | 0.618 |
| Widowed | 12.446 | 14.029 | 0.887 | 0.375 |
| States and territories | | | | |
| New South Wales | -35.850 | 29.684 | -1.208 | 0.227 |
| Victoria | -49.231 | 29.831 | -1.650 | 0.099 |
| Queensland | -1.687 | 30.139 | -0.056 | 0.955 |
| South Australia | -22.617 | 31.006 | -0.729 | 0.466 |
| Tasmania | -83.980 | 34.638 | -2.424 | 0.015 |
| Northern Territory | -41.328 | 74.694 | -0.553 | 0.580 |
| Western Australia | -28.290 | 31.224 | -0.906 | 0.365 |
| Living with dementia | 139.108 | 7.087 | 19.629 | 0.000 |
| ACFI score | | | | |
| ADL | -0.340 | 0.231 | -1.471 | 0.141 |
| BEH | 3.833 | 0.174 | 21.974 | 0.000 |
| Complex health care status | | | | |
| Low level | -287.211 | 10.512 | -27.323 | 0.000 |
| Medium level | -110.931 | 8.170 | -13.578 | 0.000 |
| Nil | -443.593 | 24.504 | -18.103 | 0.000 |

**Abbreviations:** LOS: length of stay; ACFI: Aged Care Funding Instrument; ADL: Activities of daily living; CHC: complex health care

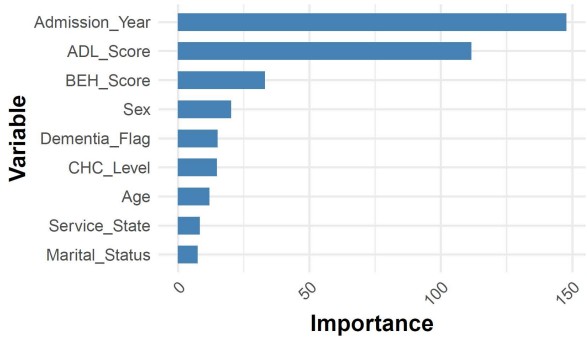

**Fig 9. Graph of random forest permutation importance.**

condition [25]. Thus, as the number of older people entering or living with dementia in RAC increases, the complexity of care and stay duration are likely to be affected as the funding allocation.

Our models are able to project dynamic average LOS for a resident with a given ACFI score, care plans should be customized to suit the individual needs. Aged-care staff should be adequately trained in managing residents with

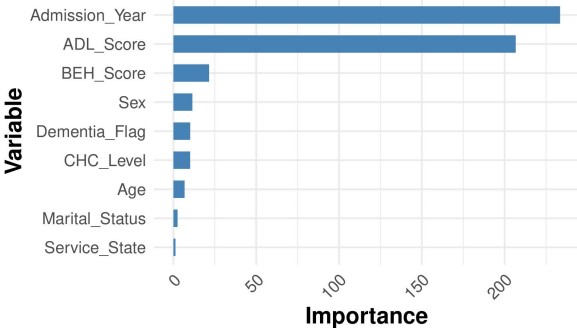

**Fig 10. Graph of gradient boosting permutation importance.**

varying degrees of behavioural issues (based on BEH score - wandering, verbal and physical aggression, and depression), which can potentially reduce residents' lengths of stay by mitigating challenges more efficiently. In recent years, issues relating to the social status, low pay and physical and emotional demand of the aged care job have been raised [26,27]. Most of the roles are filled by women which is an important factor when policymakers are considering change to aged care services such as offering better pay and training programs, to enhance the quality of care for older residents.

The superior predictive performance of ML techniques compared with the traditional linear regression is consistent with previous research that found the application of ML enhanced accuracy in the prediction of disease diagnosis, mortality and risk of readmission which are important for treatment decisions, quality of patient care and resource planning [28,29]. The application of ML algorithms to identify predictors of future health services use has not been widely explored. Previous studies using ML techniques were largely based on clinical or administrative data applied to various health conditions such as mental illness [28], neurological disorders [30] and the risk of readmission [31,32]. This is important as predictive models that are built using a broader range of predictors are more beneficial for planning targeted programs.

Our study showed that the profile of older people using RAC are also changing with a greater number of people aged 65–74 years and 90 or above accessing RAC than the other age groups. More women were more likely to have longer LOS which suggests the importance of ensuring that people working in aged care are well-prepared to meet the specific needs of an aging population, particularly women. There are also variations in the RAC use among different subpopulation such as the CALD and those living in the rural and remote areas which RAC needs to address.

While some studies have shown performance benefit of using ML over traditional statistical approaches, ML can create a black box, in which the relationships between input factors (i.e., predictors) and healthcare consumption (i.e., the outcome) are hard to interpret [33]. In contrast, traditional logistic regression analysis is widely used as it is more straightforward and can clearly articulate the contribution of different factors to the prediction outcome in the model. This transparency is vital in healthcare decision-making efforts to allow policy-makers to understand and trust the model's predictions.

In conclusion, our study identified strong predictors for LOS using ML which are important for resource decision making. The findings provided more robust evidence on the link between dementia and LOS. Older people living with dementia are an important sub-group within residential care settings that have complex and pressing care needs which would affect their duration of stay in RAC and funding decision.

## Supporting information

**S1 Table. ACFI domains and their constituent characteristics.**
(DOCX)

**S2 Table. Model comparison summary.**
(DOCX)

## Author contributions

**Conceptualization:** Jinhui Zhang, Yanlin Shi.

**Data curation:** Jinhui Zhang.

**Formal analysis:** Alexander Fracalossi, Jinhui Zhang, Yanlin Shi.

**Investigation:** Jinhui Zhang.

**Methodology:** Alexander Fracalossi, Jinhui Zhang, Yanlin Shi.

**Project administration:** Jinhui Zhang.

**Writing – original draft:** Alexander Fracalossi, Jinhui Zhang, Evelyn Lee, Yanlin Shi.

**Writing – review & editing:** Jinhui Zhang, Evelyn Lee, Yanlin Shi.

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
