## [Editor Report · Decision Letter 0]

30 Dec 2024

PONE-D-24-53339Expected Length of Stay in Residential Aged Care Facilities in Australia: Assessing the Impact of Dementia using Machine Learning

PLOS ONE

Dear Dr. Lee,

Thank you for submitting your manuscript to PLOS ONE. After careful consideration, we feel that it has merit but does not fully meet PLOS ONE’s publication criteria as it currently stands. Therefore, we invite you to submit a revised version of the manuscript that addresses the points raised during the review process.

Here are my comments and concerns.

The paper discusses modeling of Long Of Stay (LOS) of elderly population in Australia.

The rule of thumb is that Machine Learning (ML) is applicable after the authors demonstrate that simple and well-interpretable classic methods do not work. Although the authors report R-squared I have to remind that this metric increases with the number of parameters and model complexity in general. This means that R-squared is not a fair metric to compare linear regression and the Random Forest regression. Since the authors aim to predict LOS another metric, such as the Root Mean Square of LOS prediction via the leave-one-out technique should be employed.

The difference between states is not addressed. Mixed model with random intercepts that represent the state-specific LOS is easy to apply, and I recommend employing this model. I wonder how the state-specific LOS can be accounted using the ML methodology.

The nonlinearity of LOS versus Age is not addressed in regression analysis. I suggest showing the cross plot (Age,LOS) and displaying linear, quadratic, and a nonparametric regression (such as ‘loess’ in R) to understand how nonlinear the relationship is. This univariate analysis must be repeated for all other predictors.

The authors must explain/interpret ‘Importance’ of the variables in Figs 4 and 5. Importance with respect to what? How to explain Importance to Medical Doctors? Why Importance scale is different in the figures?

Data are not available: “No - some restrictions will apply.” It’s unclear why de-identified data cannot be accessed. I have to remind the authors that the data availability is a major condition for research transparency and results reproducibility.

We look forward to receiving your revised manuscript.

Kind regards,

Eugene Demidenko, Ph.D.

Academic Editor

PLOS ONE

Journal Requirements:

4. Please amend your manuscript to include your abstract after the title page.

6. Please include your tables as part of your main manuscript and remove the individual files. Please note that supplementary tables (should remain/ be uploaded) as separate "supporting information" files.

Additional Editor Comments: 

The paper discusses modeling of Long Of Stay (LOS) of elderly population in Australia.

The rule of thumb is that Machine Learning (ML) is applicable after the authors demonstrate that simple and well-interpretable classic methods do not work. Although the authors report R-squared I have to remind that this metric increases with the number of parameters and model complexity in general. This means that R-squared is not a fair metric to compare linear regression and the Random Forest regression. Since the authors aim to predict LOS another metric, such as the Root Mean Square of LOS prediction via the leave-one-out technique should be employed.

The difference between states is not addressed. Mixed model with random intercepts that represent the state-specific LOS is easy to apply, and I recommend employing this model. I wonder how the state-specific LOS can be accounted using the ML methodology.

The nonlinearity of LOS versus Age is not addressed in regression analysis. I suggest showing the cross plot (Age,LOS) and displaying linear, quadratic, and a nonparametric regression (such as ‘loess’ in R) to understand how nonlinear the relationship is. This univariate analysis must be repeated for all other predictors.

The authors must explain/interpret ‘Importance’ of the variables in Figs 4 and 5. Importance with respect to what? How to explain Importance to Medical Doctors? Why Importance scale is different in the figures?

Data are not available: “No - some restrictions will apply.” It’s unclear why de-identified data cannot be accessed. I have to remind the authors that the data availability is a major condition for research transparency and results reproducibility.

---

## [Author Response · Author response to Decision Letter 0]

11 Jan 2025

The paper discusses modelling of Length of Stay (LOS) of elderly population in Australia.

1. The rule of thumb is that Machine Learning (ML) is applicable after the authors demonstrate that simple and well-interpretable classic methods do not work. Although the authors report R-squared I have to remind that this metric increases with the number of parameters and model complexity in general. This means that R-squared is not a fair metric to compare linear regression and the Random Forest regression. Since the authors aim to predict LOS another metric, such as the Root Mean Square of LOS prediction via the leave-one-out technique should be employed.

Response: Thank you very much for your thoughtful feedback. We did indeed use cross-validation to determine the best-performing model. The R2 value was included solely for in-sample comparison, as it may be of interest to readers wishing to understand the proportion of variation explained by the model, specifically for the linear regression. However, the final model selection was based on the Root Mean Square Error (RMSE) derived from a 3-fold cross-validation process to ensure a rigorous assessment of model performance. To further clarify this point, we have updated the first paragraph of Section 3.2 accordingly.

2. The difference between states is not addressed. Mixed model with random intercepts that represent the state-specific LOS is easy to apply, and I recommend employing this model. I wonder how the state-specific LOS can be accounted using the ML methodology.

Response: Thank you for your insightful feedback and suggestions.

As illustrated in Figure 3, Table 1, and also elaborated in the Discussion section of the paper, there were no distinctive differences among states in terms of age and sex distributions or the average length of stay (LOS). This findings is also shown in Table 2 where the effect of state/territories was not statistically significant at the 1% level.

Regarding the use of machine learning models, the state/territories variable has been incorporated as a categorical feature. The results aligned closely with our expectations and the linear regression outcomes, showing contribution of state/territories importance remains minimal and close to zero.

The nonlinearity of LOS versus Age is not addressed in regression analysis. I suggest showing the cross plot (Age,LOS) and displaying linear, quadratic, and a nonparametric regression (such as ‘loess’ in R) to understand how nonlinear the relationship is. This univariate analysis must be repeated for all other predictors.

Response: Thank you for your thoughtful feedback and constructive suggestions.

In the revised manuscript, we have now included figures depicting the average length of stay (LOS) versus year, age, Activities of daily living (ADL), and Behavioral (BEH) need scores in the revised paper (please refer to Figures 6-8). The curves presented in these figures utilize the LOWESS smoothing method, as you recommended, and suggest potential non-linearity in these relationships.

To address this observation more rigorously, we have updated the linear regression model by incorporating natural splines for year, age, ADL, and BEH scores. This allows for greater flexibility in capturing non-linear patterns while maintaining interpretability. Please note that the remaining variables in the model are categorical and have been modelled using indicator variables. The differences in average LOS across the levels of these categorical variables are already presented in Table 1, making additional graphical representation redundant for this purpose.

Furthermore, Table 2 has been updated to reflect the results from the revised model. For the four continuous variables mentioned, significance tests have been provided for the overall effect of the spline terms. However, the coefficients for individual spline components are not reported, as they lack straightforward interpretability.

3. The authors must explain/interpret ‘Importance’ of the variables in Figs 4 and 5. Importance with respect to what? How to explain Importance to Medical Doctors? Why Importance scale is different in the figures?

Response: We have clarified in the revised paper that the term "importance" presented in Figures 4 and 5 were specifically referring to permutation importance, a widely used model-agnostic technique in machine learning for assessing the contribution of individual variables in predictive models.

It measures the effect of shuffling a feature on the model’s performance specifically the process involves training the model on the original data, computing the performance metric (e.g., RMSE), and then permuting each feature one at a time. After permuting a feature, the performance metric is recomputed, and the importance of the feature is calculated as the difference between the original and permuted performance scores. The formula for the importance of a feature is:

Importance of feature=Performance (original)−Performance (permuted)

The larger the drop in performance after permuting a feature, the more important that feature is for the model's predictions.

To make it clearer, we have added additional clarification regarding the concept of permutation importance in Section 3.1 in the revised manuscript to ensure greater clarity for readers.

4. Data are not available: “No - some restrictions will apply.” It’s unclear why de-identified data cannot be accessed. I have to remind the authors that the data availability is a major condition for research transparency and results reproducibility.

Response: The dataset used in this study consists of individual records organized and maintained by the Australian Institute of Health and Welfare (AIHW). Access to this dataset is restricted and granted only to approved individuals to ensure privacy and data security. Interested users may contact the AIHW directly to inquire about obtaining access to the de-identified data under the appropriate permissions.

---

## [Decision Letter · Decision Letter 1]

23 Feb 2025

PONE-D-24-53339R1Expected Length of Stay in Residential Aged Care Facilities in Australia: Assessing the Impact of Dementia using Machine LearningPLOS ONE

Dear Dr. Lee,

Thank you for submitting your manuscript to PLOS ONE. After careful consideration, we feel that it has merit but does not fully meet PLOS ONE’s publication criteria as it currently stands. Therefore, we invite you to submit a revised version of the manuscript that addresses the points raised during the review process.

New graphs in Fig. 6-8 reveal a problem with this work. 1. Why in Fig. 6 Y=Average LOS has discrete values and in Figs 7-8 close to continuous?2. Variables ADL Score and BEH Score have a weak relationship with the Y variable. It looks like the regression line is horizontal. The Y and the BEH score can be modeled as a stepwise function. All other variables presented in Fig. 6 are not significant. 3. As far as the modeling of Y is concerned, the yellow line (a spline?) is the answer with R-squared hard to beat. I suspect that a quadratic function of Year on the log scale of Y can be as good as the yellow line. In summary, it looks like ML is a wrong approach to modeling Average LOS.

We look forward to receiving your revised manuscript.

Kind regards,

Eugene Demidenko, Ph.D.

Academic Editor

PLOS ONE

Reviewers' comments:

Reviewer's Responses to Questions

**Comments to the Author**

1. If the authors have adequately addressed your comments raised in a previous round of review and you feel that this manuscript is now acceptable for publication, you may indicate that here to bypass the “Comments to the Author” section, enter your conflict of interest statement in the “Confidential to Editor” section, and submit your "Accept" recommendation.

Reviewer #1: All comments have been addressed

2. Is the manuscript technically sound, and do the data support the conclusions?

Reviewer #1: Yes

3. Has the statistical analysis been performed appropriately and rigorously? 

Reviewer #1: Yes

4. Have the authors made all data underlying the findings in their manuscript fully available?

Reviewer #1: Yes

5. Is the manuscript presented in an intelligible fashion and written in standard English?

Reviewer #1: Yes

6. Review Comments to the Author

Reviewer #1: The authors compared linear regression and machine learning for predicting the length of stay (LOS) in residential aged care facilities. They identified dementia as an important factor and confirmed that machine learning outperformed linear regression in prediction accuracy. While this is an interesting study, several flaws should be addressed.

Abstract

The authors should present data rather than conclusions in the results section.

The statement: “On average, residents with dementia stayed 6 months longer compared to those without, which suggests a positive association between LOS and levels of complex health care (CHC).” is inappropriate for the results section. Instead, present the proportion of residents with dementia staying less than and more than six months, along with the corresponding p-value.

The claim: “Owing to the presence of heteroscedasticity and non-normality of LOS, the application of linear regression is likely to skew the estimates. Our study showed that among the different ML techniques, GBDT provided better predictive performance as it could capture nonlinear relationships and was less sensitive to changes or dependent on stringent assumptions compared with traditional linear regression.” should be supported with specific results. Present the performance metrics of different models, such as AUC for categorical outcomes and R² for continuous outcomes (e.g., DOI: 10.1038/s41598-024-82615-0).

Introduction

The authors extensively discuss the significance of aged care. However, given that the study focuses on machine learning for LOS prediction, more discussion on AI techniques is warranted. Consider incorporating relevant references (e.g., DOI: 10.69854/jcq.2024.0011).

Methods

Cite relevant references for the machine learning models used, such as Random Forest, Linear Regression, and Gradient Boosting.

Move the Training and Validation section after the introduction of models for better flow.

Results

Major findings should be presented with specific data points, allowing readers to understand the results more clearly.

Discussion

The discussion section should be separate from the results section and structured as 4.0 Discussion to improve readability and clarity.

7. PLOS authors have the option to publish the peer review history of their article (what does this mean? ). If published, this will include your full peer review and any attached files.

**Do you want your identity to be public for this peer review?** For information about this choice, including consent withdrawal, please see our Privacy Policy .

Reviewer #1: No

---

## [Author Response · Author response to Decision Letter 1]

28 Feb 2025

Overall response:

We thank the reviewer for the comments, and we apologise for the misleading conclusions that were actually drawn from our figures in the previous submission. The purpose of presenting Figures 6-8 in the earlier manuscript was to illustrate the relationship between LOS (Y variable) and explanatory variables (i.e., Year, BEH and ADL scores).

Due to our large sample size (n=509,142), we realise that it is not efficient or possible to show the complete scatter plot with the full dataset. As such, in the previously figures, we plotted an average LOS for each unique value of an explanatory variable. The LOWESS curve was applied to the resulting average LOS shown in each figure, and these curves are not necessarily consistent with that could be obtained using the complete dataset.

We have now clarified and reproduced our four plots in the current revised manuscript. In each plot, the LOWESS curve is obtained using the complete dataset, and the scatter points consist of 5,000 randomly selected subsample from the complete dataset. These scatter points should only be used as a snapshot of the full sample, and the non-linear patterns shown by the LOWESS curves are more important. In these new figures, it is clear that each of the four continuous variables may non-linearly influence the LOS. The paper has been revised to highlight these updates.

In addition, we would like to restate our logic and steps of our analyses:

1. Provide plots of LOS vs explanatory variables to preliminarily analyse the potential relationships.

2. The observed non-linear relationships preliminarily support the potential usefulness of ML models, which can accommodate such non-linearity without having to specify it. In comparison, the four continuous variables are modelled via splines in the usual linear regression to accommodate non-linearity.

3. Both linear regression and ML models support the significance of ADL and BEH scores.

4. ML models outperforms the linear regression as evidenced by RMSE and (pseudo) R-square, and thus is recommended as the final models to study and project LOS in our present study.

We believe that these new results can effectively address the previous concerns. Therefore, the three comments may no longer be applicable to our revised paper. Nevertheless, we decided to provide point-to-point response for the completeness.

1. Why in Fig. 6 Y=Average LOS has discrete values and in Figs 7-8 close to continuous?

Response: Thank you for your comments. In these figures, we plotted the average LOS for each unique value of the corresponding explanatory variables. Discrete values were found for Year since there were only 13 distinct values. In contrast, the other explanatory variables (ADL and BEH scores in Figures 7 and 8, respectively) had many more unique values and thus appeared to be continuous data.

2. Variables ADL Score and BEH Score have a weak relationship with the Y variable. It looks like the regression line is horizontal. The Y and the BEH score can be modeled as a stepwise function. All other variables presented in Fig. 6 are not significant.

Response: We thank the reviewer for highlighting this. As outlined above, these figures showed the LOWESS curve of unique values of each explanatory variable against the corresponding average LOS. They do not represent the true curve that may be obtained using the full sample. To more systematically justify the significance, the p-value in linear regression and importance scores in ML models should be used instead. In both cases, the ADL Score and BEH Score are believed to be significant and/or among the most important explanatory variables.

3. As far as the modeling of Y is concerned, the yellow line (a spline?) is the answer with R-squared hard to beat. I suspect that a quadratic function of Year on the log scale of Y can be as good as the yellow line.

Response: Thank you for your comment. While a quadratic model might seem to capture the non-linearity observed in the scatter plot, we chose to use splines in linear regression because it is more general and flexible. Splines allow the relationship to vary across different ranges of the explanatory variable, without imposing the fixed shape that a quadratic function does. This flexibility is especially useful if the true relationship deviates from a simple quadratic form, especially when other explanatory variables are later included in the full model.

Additionally, the ML models employed are designed to capture complex non-linear patterns without the need to specify a particular functional form in advance. Their ability to learn directly from the data means that they can adapt to various underlying relationships, including but not limited to linear, quadratic and other patterns.

In other words, our approaches (spline in linear regression and ML models) minimize the risk of model misspecification and allow for a better fit if the true relationship is more complex than a simple quadratic. More importantly, as evidenced by (pseudo) R-square and RMSE, ML models outperform the linear regression. This validates the reason that they are chosen as the final model in this paper.

---

## [Editor Report · Decision Letter 2]

9 Mar 2025

Expected Length of Stay in Residential Aged Care Facilities in Australia: Assessing the Impact of Dementia using Machine Learning

PONE-D-24-53339R2

Dear Dr. Lee,

We’re pleased to inform you that your manuscript has been judged scientifically suitable for publication and will be formally accepted for publication once it meets all outstanding technical requirements.

As a comment regarding your graphs 1a-1d, I suggest showing vertical kernel densities of Days's distribution. You can adopt Figures 8 and 9 from my book Demidenko E (2020). Advanced Statistics with Applications in R on page 677 and 678.

Kind regards,

Eugene Demidenko, Ph.D.

Academic Editor

PLOS ONE

---

## [Editor Report · Acceptance letter]

PONE-D-24-53339R2

PLOS ONE

Dear Dr. Lee,

I'm pleased to inform you that your manuscript has been deemed suitable for publication in PLOS ONE. Congratulations! Your manuscript is now being handed over to our production team.

Kind regards,

on behalf of

Dr. Eugene Demidenko

Academic Editor

PLOS ONE